# Building a Knowledge Graph for Products and Solutions in the Automation Industry

Thorsten Liebig[2][0000−0002−2810−7315], Andreas Maisenbacher[1], Michael Opitz[2], Jan R. Seyler[1], Gunther Sudra[1], and Jens Wissmann[1][0000−0001−6434−1355]

[1] Festo AG & Co. KG, Ruiter Str. 82, 73734 Esslingen, Germany
`forename.surname@festo.com`
[2] derivo GmbH, Münchner Str. 1, 89073 Ulm, Germany
`surname@derivo.de`

**Abstract.** This paper reports on an ongoing Knowledge Graph generation project at the industrial automation company Festo. The project aims at establishing a well defined and easily traceable extract, transform, load (ETL) process of technical data on automation products into Festo's semantic data platform (FSP) to serve complex configuration tasks. Starting with a brief illustration of the data sources and requirements we describe a staged Knowledge Graph generation process that draws on common data extraction techniques as well as on specific semantic transformation methods such as the mapping framework R2RML or ontology reasoning. We will report on the project status as well as discuss our design decisions towards a reusable FSP Knowledge Graph data ingestion process.

**Keywords:** Knowledge Graph · R2RML · ontology reasoning.

## 1    Introduction

Festo is a multinational industrial control and automation company based in Germany. The Festo Semantic Platform (*FSP*) is a major initiative at Festo to create and provide a Knowledge Graph and reasoning-based services about products and solutions for Festo's core business: factory automation. For instance, tasks such as the computation of compatibility of product components within electrical drive trains is provided by the FSP infrastructure using ontology reasoning and SPARQL querying [1].

### 1.1    Current Solution and its Challenges

Our current solution is structured as an ETL process. We use this approach as our data undergo a series of sequential processing and refinement steps that include materialisation of reasoning results and addition of expert data before the final result is delivered. Ontology based data access (OBDA) [7] is currently not our focus but might complement our approach in future, e.g. for initial database exploration. Although our processing is performed stepwise, the logic of the FSP

is currently implemented as a monolithic Java application. The application uses SQL to retrieve data from relational databases, Java code to classify the data and to create unique identifiers (IRIs), and the OWL API [3] to transform the results into a Knowledge Graph so that it conforms to its respective ontology schema. Experiences with this ETL solution within the last two years have revealed drawbacks with respect to source data changes and maintainability of the transformation logic that caused significant data analysis efforts for the FSP team. The current solution is difficult to debug, as it is hard to identify whether a Knowledge Graph change or error is caused by a source data update, a bug in the encoding of the data enrichment or transformation logic, or a flaw in the ontology schema. Furthermore, the monolithic design is not easy to understand and to maintain since it consists of multiple processing tasks that are composed into one piece of code.

To better deal with this issues and to establish a well defined Knowledge Graph ETL process, Festo has recently started an initiative to improve the Knowledge Graph generation. The following list summarizes the requirements for the FSP Knowledge Graph generation pipeline currently in progress:

**Change detection and propagation:** The FSP services rely on data that is typically distributed over various source databases that are managed by processes that vary in their level of reliability and their update cycles. In the general use case of technical product data, the ETL process has to integrate data sources of different quality. There is well maintained data from SAP about products that have reached a certain development maturity or are already on sale. In addition there is data about upcoming products from a database maintained by product development. Data in the latter database may change without any traceable history or notification of any of its data consumers. Therefore, it's a key requirement for our Knowledge Graph generation pipeline to incorporate a data change detection and tracing mechanisms that can detect data changes as early as possible in the data sources to support Knowledge Graph debugging.

**Tracking and logging:** The FSP Knowledge Graph generation should follow a step wise processing pipeline determined by its functional demands and with a clear specification of its processing blocks and data interfaces. Furthermore, each of the processing blocks has to report about its outcome, i.e., whether it has terminated successfully, and provide statistics on the results. Ideally there is a test and evaluation step after each of the processing blocks that checks for unexpected results. All the reporting should be accessible in a way that supports building a dashboard that allows to operate and monitor the ETL process to discover potential problems as early as possible.

**Declarative approach:** Since the FSP is committed to Semantic Technologies, the ETL logic should be as declarative as possible instead of being buried somewhere in program code. In other words, we prefer to have data classifi-

cation done by ontological rules and reasoning as well as data transformation based on mapping rules rather than by procedures, table functions, etc.

**Based on standards:** Whenever possible the Knowledge Graph generation pipeline should employ standards that have shown to work in industry, are widely adapted in the community and for which tools and engines are available. This refers in particular to the data classification and transformation part. Since the FSP already uses OWL 2 RL reasoning and SWRL, the ETL process should adapt to this fragment and syntax formats. For the transformation part, we plan to evaluate mappings and tools from relational database tables to RDF datasets such as R2RML or RML.

## 2   Data Processing Pipeline

Our target data processing pipeline consists of several processing stages that are performed on a microservice-oriented architecture as shown in Figure 1. In the following we describe the stages of this process.

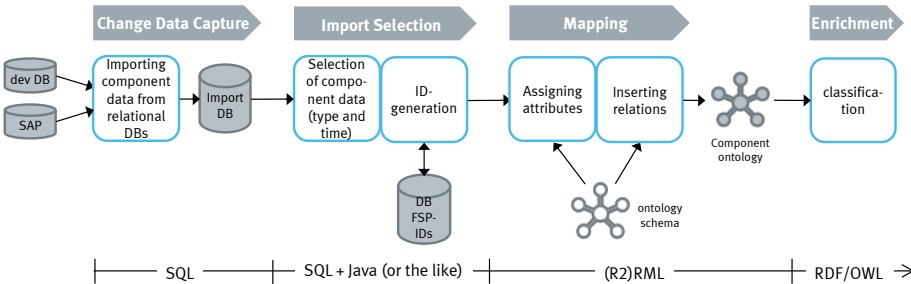

**Fig. 1.** Overview of the Knowledge Graph ETL pipeline.

The pipeline is in charge of transforming product data as well as data on complex automation systems into a Knowledge Graph for a given ontology schema. A typical product in the product domain is an electric axis. An axis can move, push, or press a work-piece. An axis furthermore is driven by a motor which in turn is actuated by a controller. The whole chain from axis to controller is called a drive train. It also incorporates a mounting kit that allows to mount the motor to the axis and optionally a gear. The ESBF axis[3] family is a such a product tho which we refer in our running example in the following.

Even if the number of products is few in numbers with some thousands the ETL process to generate a Knowledge Graph of high quality is a business critical element since it produces the source data of subsequent semantic processing steps. As an example, based on the basic products and data about technical compatibility we compute all nearly 100 million technically valid drive trains.

---

[3] https://www.festo.com/cat/en-gb_gb/data/doc_ENGB/PDF/EN/ESBF_EN.PDF

## 2.1   Change Data Capture

Due to the fact that one of our main data sources is an un-revisioned database, we have to deal with new, deleted or updated product data during Knowledge Graph ingestion. We address this during the Change Data Capture phase.

Technically, this is delivered by a layer of SQL databases, called "Import DB" in Fig. 1, that are in our sovereignty. We employ type 2 of the Slowly Changing Dimensions (SCD) method known from Data Warehousing [5] to detect changes and to historicize the data. SCD type 2 tracks historical data by maintaining previous versions of source data records by adding so called surrogate keys. A surrogate key is a unique identifier not derived from the application data itself but introduced for technical reasons only. For the purpose of tracking data changes, we generate a composite surrogate key that consists of a product specific identifier (`ID`), and two time stamps (`VALID-FROM` and `VALID-UNTIL`). The time stamps specify the time interval the particular product data record was detected in the source database. The most current data record of a product is the one that has the predefined surrogate high date for `VALID-UNTIL` (`2099-01-01` in our case).

On request of the processing pipeline the Change Data Capture stage starts checking for updates in the source databases by comparing with the current data records in the import database. The latter will receive new rows for any detected change to reflect the historical state of the source database.

| ID | VALID-FROM | VALID-UNTIL | TYPECODE | MAX_VELOC | MAX_LOAD | ACCUR |
|---|---|---|---|---|---|---|
| 1 | 2019-01-08 | 2019-02-09 | ESBF-BS-32-%%-5P | 0.55 | 100.0 | 0.02 |
| 1 | 2019-02-10 | 2099-01-01 | ESBF-BS-32-%%-5P | 0.55 | 100.0 | 0.01 |
| 2 | 2019-01-01 | 2099-01-01 | ESBF-LS-32-%%-2.5P | 0.05 | 60.0 | 0.05 |
| 3 | 2019-01-01 | 2099-01-01 | ESBF-LS-32-%%-2.5P-S1 | 0.05 | 60.0 | 0.05 |

Table 1. SCD type 2 table for axes with validity times and ID as surrogate keys

Table 1 depicts an example of our import database for axes. It contains some sample axes of the ESBF family and a fraction of its technical data. For the axis `ESBF-BS-32-%%-5P` (with `ID` 1) the import database has two entries which account for a data update. More precisely, the axis accuracy (last column labeled `ACCUR`) changed on February $10^{\text{th}}$ to the new value "`0.01`". The corresponding row represents the current valid data record for this axis since it holds the predefined surrogate high date in `VALID-UNTIL`.

In addition to capturing data changes by applying the SCD method, the import database allows us to presort the data. As a result, the import database follows the design principle, that every table corresponds to a product category that is also represented by a class in our ontology schema about drive trains. Therefore, every row in the table is designed with the intention to represent an individual with its technical data properties in the Knowledge Graph. Other tables represent information about relationships between individuals.

To this end, the import database serves two main purposes. It allows to track changes in our source data on the one hand, and on the other hand it makes

the translation to a Knowledge Graph easier since it's a step towards a graph of individuals and relationships.

## 2.2  Import Selection

The main task of this stage is to select product data from the import database and to add unique identifiers to this data in preparation for a mapping into a Knowledge Graph format.

When considering an initial full load, this stage selects the most current records from the import database. These records are then aggregated into tables such that these tables roughly correspond to product classes of the ontology schema. In the domain of electric drive trains these are axis, mounting kit, gear, motor, or controller plus accessories etc. Due to the history of data changes in the import database of the previous step, the ETL process can also select previous data versions of particular products. For instance, it allows to select the latest data for all products except for axes for which we can choose to have data from a particular point in time. As mentioned above, this is key to generate reasonable Knowledge Graphs from data with varying level of maturity.

| ID | TYPECODE | MAX_VELOC | MAX_LOAD | ACCUR |
|---|---|---|---|---|
| A000576 | ESBF-BS-32-%%-5P | 0.55 | 100.0 | 0.01 |
| A000628 | ESBF-LS-32-%%-2.5P | 0.05 | 60.0 | 0.05 |
| A000629 | ESBF-LS-32-%%-2.5P-S1 | 0.05 | 60.0 | 0.05 |

Table 2. Selection table of axis data with IDs

For each of these domain objects a unique identifier is required. Such an identifier is important since Knowledge Graphs need a key to distinguish instances from each other. The management of identifiers is provided by an FSP service. For electric drives the FIDGET (Festo ID Generator) service is the central authority for looking up an existing or generating a new identifier. Typically, an identifier is computed from a set of technical characteristics (basic products) or its components in case of a complex system (e.g. a 3D portal). To facilitate debugging the FIDGET service keeps its identifier constant over time. Therefore, we can easily compare Knowledge Graphs generated at different times to check for changes on component level. Table 2 depicts an import selection from the import database with the latest technical data and identifiers obtained from the ID service.

## 2.3  Data Mapping

The mapping step involves the transformation of relational data into instances of the Knowledge Graph, instance attributes and relations between those instances. The target format for the FSP component ontology is any valid syntax specified by W3C for RDF or OWL. The mapping step is deliberately designed to be accomplished by a pure declarative transformation not incorporating any

procedural data manipulation parts. This is possible because of the preceding import selection step, that has aggregated and aligned the source data appropriately as well as added unique identifiers suitable to serve as IRIs. As a result, this step can be carried out by a R2RML mapping specification executed by an R2RML engine. This allows to purely draw on W3C's R2RML which hopefully provides flexibility in the selection of tools and engines to choose the optimal deployment option with respect to performance, data load, etc.

The following exemplary R2RML mapping shows how technical data is transformed into a RDF-based Knowledge Graph:

```
@prefix rr: <http://www.w3.org/ns/r2rml#>.
@prefix fsp: <http://www.festo.com/edrive#>.
@prefix xsd:  <http://www.w3.org/2001/XMLSchema#> .
@prefix rdf:  <http://www.w3.org/1999/02/22-rdf-syntax-ns#> .
@prefix rdfs: <http://www.w3.org/2000/01/rdf-schema#> .

<#TriplesMap1>
  rr:logicalTable [ rr:tableName "import-selection" ];
    rr:subjectMap [
      rr:template "http://www.festo.com/edrive#{ID}";
      rr:class fsp:Axis;];
    rr:predicateObjectMap [
      rr:predicate rdfs:label ;
      rr:objectMap [ rr:column "TYPECODE" ]; ];
    rr:predicateObjectMap [
      rr:predicate fsp:max-velocity ;
      rr:objectMap [ rr:column "MAX_VELOC"; rr:datatype xsd:double ]; ];
    rr:predicateObjectMap [
      rr:predicate fsp:max-loading ;
      rr:objectMap [ rr:column "MAX_LOAD"; rr:datatype xsd:double ]; ];
    rr:predicateObjectMap [
      rr:predicate fsp:repetition-accuracy ;
      rr:objectMap [ rr:column "ACCUR"; rr:datatype xsd:double ]; ].

<#TriplesMap2>
  rr:logicalTable [ rr:sqlQuery
          """SELECT ID FROM import-selection
              WHERE REGEXP_LIKE(TYPECODE, '.*-S1.*') = '1'""" ];
    rr:subjectMap [ rr:template "http://www.festo.com/edrive#{ID}"; ];
    rr:predicateObjectMap [
      rr:predicate rdf:type;
      rr:objectMap [ rr:template "http://www.festo.com/edrive#IP65" ]; ].
```

The first part of the mapping simply associates a label as well as technical data to a Knowledge Graph instance that is identified by an IRI compiled from the given identifier. The second part is an example of a typecode based instance classification. The latter mapping checks for a particular typecode fragment ("S1") that indicates a particular protection level (IP65: dust and water protected). The typecode chart for ESBF axes is shown in Fig. 2.

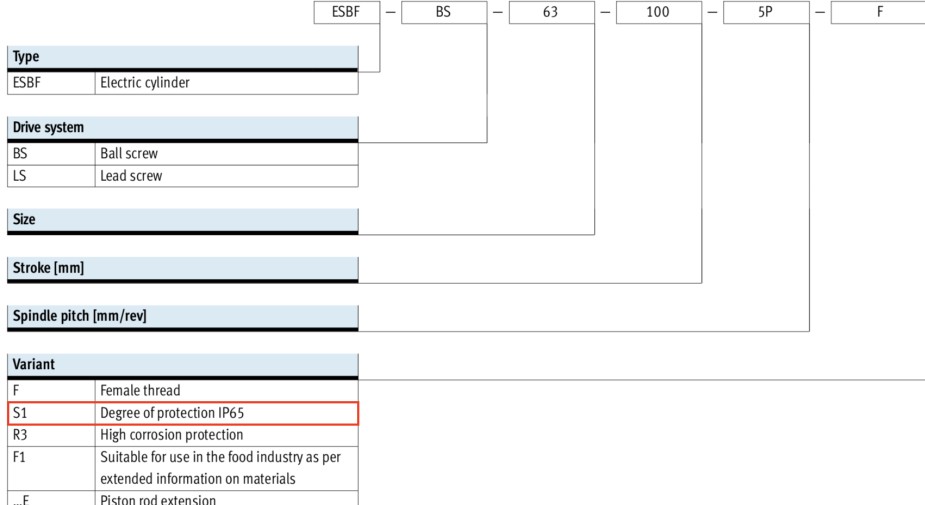

| ESBF | – | BS | – | 63 | – | 100 | – | 5P | – | F |

**Type**

| ESBF | Electric cylinder |
| --- | --- |

**Drive system**

| BS | Ball screw |
| --- | --- |
| LS | Lead screw |

**Size**

**Stroke [mm]**

**Spindle pitch [mm/rev]**

**Variant**

| F | Female thread |
| --- | --- |
| S1 | Degree of protection IP65 |
| R3 | High corrosion protection |
| F1 | Suitable for use in the food industry as per extended information on materials |
| ...E | Piston rod extension |

**Fig. 2.** ESBF typecode chart

All Festo products are classified according to typecodes. Not all of these are relevant within the applications supported by the FSP Knowledge Graph. The R2RML mapping therefore just takes care of those which are required for later reasoning tasks.

In a brief analysis we evaluated the R2RML resp. RML editors KARMA [6], RML Editor [2], and Map-On [9] most notably with respect to mapping expressivity, maturity, and their license model. It turned out, that most of the editors seem to do the job in principle. Some provide extensions such as Python scripting or importing various other data formats that allow much more of what is required in this step of our Knowledge Graph pipeline. Result of the mapping in our running example is the following:

```
@prefix fsp: <http://www.festo.com/edrive#> .
@prefix rdfs: <http://www.w3.org/2000/01/rdf-schema#> .
@prefix xsd: <http://www.w3.org/2001/XMLSchema#> .

fsp:A000576 a fsp:Axis ;
        rdfs:label "ESBF-BS-32-%%-5P" ;
        fsp:max-velocity "0.55"^^xsd:double ;
        fsp:max-loading "100.0"^^xsd:double ;
        fsp:repetition-accuracy "0.01"^^xsd:double .

fsp:A000628 a fsp:Axis ;
        rdfs:label "ESBF-LS-32-%%-2.5P" ;
        fsp:max-velocity "0.05"^^xsd:double ;
        fsp:max-loading "60.0"^^xsd:double ;
        fsp:repetition-accuracy "0.05"^^xsd:double .
```

```
fsp:A000629 a fsp:Axis, fsp:IP65 ;
        rdfs:label "ESBF-LS-32-%%-2.5P-S1" ;
        fsp:max-velocity "0.05"^^xsd:double ;
        fsp:max-loading "60.0"^^xsd:double ;
        fsp:repetition-accuracy "0.05"^^xsd:double .
```

As mentioned above we are interested in applying a standard mapping approach not making use of non-standard extensions such as procedures etc. Ideally, we want to specify the business logic fully within the mapping language. Further processing should either in the previous or following steps. For those cases where R2RML does not provide enough mapping expressivity we decided to use SWRL in the post-R2RML step (see next section).

Although none of the evaluated (R2)RML editors failed for our purpose we are not using one of them. Instead we are currently add annotations to your ontology schema that allows us to generate the R2RML mapping above via transformation from these annotations. The huge advantage of this approach is that it incorporates this information into the key place of metadata about the Knowledge Graph, namely the ontology schema.

The engine should then be able to run the mapping specification independently from front ends in order to be able to execute it from a workflow engine.

### 2.4   Data Enrichment

The goal of the enrichment steps is to enhance the Knowledge Graph by making implicit information explicitly available with the help of OWL reasoning.

A key task in this respect is the categorization of Festo components according to product families or technical characteristics that are relevant for the subsequent services. Some classification tasks are already done at the mapping stage based on a typecode decomposition with the help of regular expressions (compare with the protection level IP65 classification above). Other categorization tasks that do not directly relate to a typecode feature. They are handled via OWL reasoning either based on OWL axioms or SWRL. For instance, components suitable for food manufacturing have to comply to different protection classes depending on their component type (a controller typically requires a lower protection class because it is less close to food in comparison to an axis). Another example relates to information that need to be propagated through the Knowledge Graph. For instance, the size of a drive train originate from the respective size of its axis which is automatically derived by an OWL axiom.

After enrichment the result data of the ETL are ready to be used in the core part of our semantic platform. For example, instance data about axes, motors, and controllers are combined with background knowledge that is described in OWL and SWRL. A reasoning step performed by RDFox [8] can than infer compatibility relationships between the products as sketched in fig. 3 on the next page. A description of the model and rules is outside the scope of this paper. Some more details are described in [1].

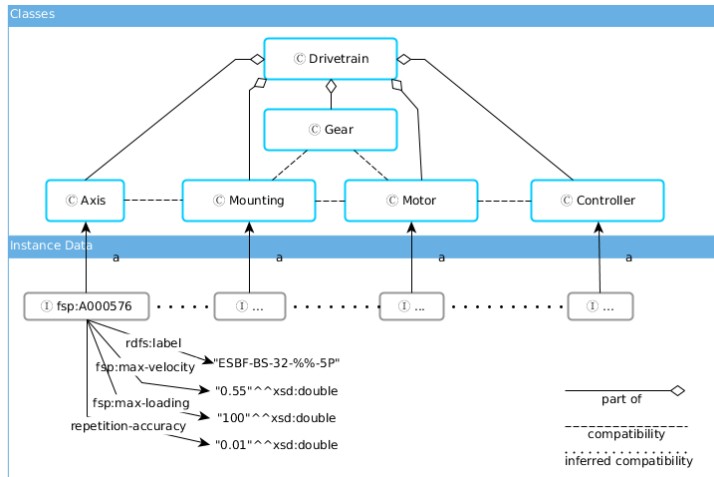

**Fig. 3.** Fragment of the resulting knowledge graph.

## 3 Infrastructure and Continuous Delivery

Our original system was conceived as a set of standalone applications. As system complexity and availability requirements grow, we need to align our system with software engineering and continuous delivery [4] practices used at Festo. Further we anticipate that future projects make it necessary to continuously evolve and restructure the processing pipeline. The main patterns we follow are introduction of a *micro service architecture* and *container virtualization*.

We use Java as programming language. For micro services there is no fixed framework. However, in our first efforts we use the OpenAPI 3 standard to specify our REST interfaces and use Swagger to generate Spring Boot service stubs. Our codebase is maintained in Git repositories. As build system we use Maven. Builds are triggered and monitored using JetBrains TeamCity. To reproduce the system environment we package the artifacts in Docker containers and use Docker Compose to run multi container applications. Maven artifacts and Docker images are deployed to the company internal Artifactory repository.

## 4 Workflow Control and Monitoring

To orchestrate the execution of the processing steps and monitor the progress we need a workflow solution.

The workflow system should allow *automated execution and scheduling* of the execution steps. This includes for example regularly checking the database for changes and trigger the mapping process. Information of the current state of processing and failure states should be easily retrievable. Processing state could for example indicated by means of graph visualization of the workflow. For other

metrics such as number of generated instances of a specific concept a dashboard could be helpful.

We currently consider Apache Airflow[4] as a platform. It provides to programmatically author, schedule and monitor workflows. It provides templates for common workflow steps such as SQL or REST calls. Custom processing steps can be implemented in Python.

## 5   Status and Outlook

In this work we described our ongoing effort to refine the data processing pipeline of the Festo Semantic Platform. The system is currently used productively at Festo.

In our target architecture, the processing pipeline is structured into four processing phases — change data capture, import selection, mapping and enrichment. We are in the process of aligning our processing artifacts to this architecture. For parts of the system we plan to evaluate the technology. We identified (R2)RML as a promising technology and started evaluating tools and implementations. However, an upfront evaluation of the overall approach remains a challenge.

The Festo Semantic Platform will evolve over time as new knowledge domains are added and further company systems are integrated. In order to gain more flexibility to restructure the processing flow when requirements change we rework our services as *microservices* and move to *container* based application management. To achieve transparency of the processing we intend to introduce a *workflow control* system and monitoring.

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
