# OpenReview forum: "Building a Knowledge Graph for Products and Solutions in the Automation Industry"
_eswc-conferences.org/ESWC/2019/Workshop/KGB — KGB 2019_

### Official Review · ~Ahmet_Soylu1 · 2019-03-25
**Building a Knowledge Graph for Products and Solutions in the Automation Industry**

**Rating:** 3
**Confidence:** 3

**Review:**

The paper reports an ETL-based knowledge graph generation process within an industrial context using some data warehousing techniques and well-known semantic technologies such as R2RML.

The process, techniques, and tools used in this work are quite typical, nothing new, I would say; nevertheless, it represents challenges and overall technical methodology in an industrial context in a compact and nice way. It also links KG up to the reasoning process for checking product compatibility, although details are not provided. Overall, the paper may lead to good discussions during the workshop.

The paper misses two important elements:

- one is the related work: The authors could describe and compare related approaches and methodologies. For example, it would have been interesting to see why authors did not go for a virtualization based approach using OBDA instead of ETL.

- the second one is evaluation: Although I do understand the authors' point that it is hard to provide an upfront evaluation at this point, some sort of mini-evaluation would have been useful. The authors should also provide some numbers on the size of data being transformed, the current and expected size of KG etc. to put things in a context.

Minor: In the abstract, the authors refer to Festo and FSB. These may not be known by every reader, I would either avoid using them or provide a very brief description in the abstract.

---

### Official Review · ~Boris_Villazon-Terrazas1 · 2019-04-01
**Description of a Knowledge Graph building process based on ETL and "(R2)RML"**

**Rating:** 3
**Confidence:** 3

**Review:**

This paper presents a Knowledge Graph building process. The process relies on ETL techniques and (R2)RML mappings.

Pros
------
The paper presents a real case scenario, in which semantic technologies are applied. For me it is important to have/follow a KG generation process, but more important is to exploit such Knowledge.
The paper identifies a set of requirements for the KG generation, describes the data processing pipeline along with infrastructure and workflow.
From the technological point of view, the KG building process is relying on microservice architecture, container ready, and automated execution and scheduling techniques.

Cons
-------
Seems to be that the authors still not fully implement (R2)RML mappings. They said they are in the process of evaluating the tools. So, how real is this application scenario for KG building process.
The authors did not compare their architecture/system with some state-of-the-art R2RML-based KG generation system, not even at high level.
The authors did not tell anything about how to evaluate/assess each one of the identified requirements, and the global KG building process. How to check they are not missing any knowledge or they are creating inconsistencies.
It is not clear for me, if the KG building process includes the construction of the schema and/or instances.
I would like to know the data volume/size (input data and KG generated elements) the authors expect to have at the beginning, and in each new iteration.

Minor comments
-----------------------
The authors should put the acronym Festo Semantic Data (FSP) at the beginning of the paper. Moreover, they should include a short description about Festo.
I would suggest to change the first requirement to "Change detection and propagation"
I would suggest to include references to R2RML and RML
I did not get this sentence: " ...This stage keeps track of data changes and aggregates technical data of *products.resp.* ignores irrelevant data ... "
Change , i.e. -> , i.e.,

I would say this paper can be the introduction to have a F2F discussions about how to design and implement a real case scenario of  R2RML-based KG construction. Moreover, it can be the starting point to identify what are the benefits from the business perspective to use KGs against other approaches.

---

### Decision · Program_Chairs · 2019-04-08
**Acceptance Decision**

**Decision:**

Accept

**Comment:**

This contribution is accepted for presentation at the KGB2019 workshop, and for inclusion in its proceedings.